# Phospholipases in Gliomas: Current Knowledge and Future Perspectives from Bench to Bedside

**DOI:** 10.3390/biom13050798

**Published:** 2023-05-07

**Authors:** Maria Vittoria Marvi, Irene Neri, Camilla Evangelisti, Giulia Ramazzotti, Sofia Asioli, Matteo Zoli, Diego Mazzatenta, Niccolò Neri, Luca Morandi, Caterina Tonon, Raffaele Lodi, Enrico Franceschi, James A. McCubrey, Pann-Ghill Suh, Lucia Manzoli, Stefano Ratti

**Affiliations:** 1Cellular Signalling Laboratory, Department of Biomedical and Neuromotor Sciences (DIBINEM), University of Bologna, 40126 Bologna, Italy; mariavittoria.marvi2@unibo.it (M.V.M.); irene.neri3@unibo.it (I.N.); camilla.evangelisti@unibo.it (C.E.); giulia.ramazzotti@unibo.it (G.R.); 2Department of Biomedical and Neuromotor Sciences (DIBINEM), University of Bologna, 40126 Bologna, Italy; sofia.asioli3@unibo.it (S.A.); matteo.zoli4@unibo.it (M.Z.); diego.mazzatenta@unibo.it (D.M.); luca.morandi2@unibo.it (L.M.); caterina.tonon@unibo.it (C.T.); raffaele.lodi@unibo.it (R.L.); 3Programma Neurochirurgia Ipofisi—Pituitary Unit, IRCCS Istituto delle Scienze Neurologiche di Bologna, 40124 Bologna, Italy; niccolo.neri2@studio.unibo.it; 4Functional and Molecular Neuroimaging Unit, IRCCS Istituto delle Scienze Neurologiche di Bologna, 40139 Bologna, Italy; 5Nervous System Medical Oncology Department, IRCCS Istituto delle Scienze Neurologiche di Bologna, 40139 Bologna, Italy; e.franceschi@isnb.it; 6Department of Microbiology and Immunology, Brody School of Medicine at East Carolina University, Greenville, NC 27858, USA; mccubreyj@ecu.edu; 7Korea Brain Research Institute (KBRI), Daegu 41062, Republic of Korea; pgsuh@kbri.re.kr; 8School of Life Sciences, Ulsan National Institute of Science and Technology (UNIST), Ulsan 44919, Republic of Korea

**Keywords:** phospholipases, brain tumors, gliomas, therapeutic target, prognostic biomarker

## Abstract

Phospholipases are essential intermediaries that work as hydrolyzing enzymes of phospholipids (PLs), which represent the most abundant species contributing to the biological membranes of nervous cells of the healthy human brain. They generate different lipid mediators, such as diacylglycerol, phosphatidic acid, lysophosphatidic acid, and arachidonic acid, representing key elements of intra- and inter-cellular signaling and being involved in the regulation of several cellular mechanisms that can promote tumor progression and aggressiveness. In this review, it is summarized the current knowledge about the role of phospholipases in brain tumor progression, focusing on low- and high-grade gliomas, representing promising prognostic or therapeutic targets in cancer therapies due to their influential roles in cell proliferation, migration, growth, and survival. A deeper understanding of the phospholipases-related signaling pathways could be necessary to pave the way for new targeted therapeutic strategies.

## 1. Introduction

The healthy human brain is composed of approximately 60% lipids distributed predominantly in myelin and white matter. Among lipids, phospholipids (PLs) represent the most abundant species contributing to the biological membranes of its component neurons [1]. It has been demonstrated a strong correlation between the phospholipid membrane composition and the fluidity, essential for maintaining the cellular and subcellular structure and the function in the brain [1]. Consequently, phospholipids and their metabolites’ alteration can affect the intrinsic properties of cell membranes, modulating critical proteins’ activation and the entire cellular signaling, often leading to pathological implications and cancer. However, the mechanisms of phospholipid remodeling in tumor cells are still poorly understood.

Phospholipids consist of a glycerol backbone with two ester-linked fatty acyl fractions at the stereospecific number-1 (*sn-*1) and the stereospecific number-1 (*sn-*2) positions and a phosphodiester-linked hydrophilic head group at the *sn-*3 position that can be represented by choline, serine, ethanolamine, or inositol [1].

Phospholipids can be hydrolyzed into bioactive lipid mediators, such as diacylglycerol, phosphatidic acid, lysophosphatidic acid, and arachidonic acid by phospholipases [2]. Phospholipases are lipolytic enzymes which have been shown to be involved in the regulation of a plethora of cellular physiological and pathological mechanisms as proliferation, survival, migration, inflammation, and tumorigenesis [3]. Phospholipases are characterized by different activities, substrate preferences, and distribution [4], and they are categorized into four major classes, termed A, B, C, and D, according to the site of cleavage within PLs (Figure 1) [1,5].

Notably, for each family, there are several isoforms with different roles and expression patterns in cell types and organelles [1]. Even if each phospholipase regulates peculiar cell signaling mechanisms, they can share common signaling molecules that work as upstream regulators or downstream effectors [3]. 

In this review, it is summarized the current knowledge about the role of phospholipases in brain tumor progression, representing promising prognostic or therapeutic targets in cancer therapies due to their influential roles in cell proliferation, migration, growth, and survival. High- and low-grade diffuse adult-type gliomas are mainly considered, which originate from the supporting neuroglial cells of the Central Nervous System (CNS). Indeed, according to the most recent World Health Organization (WHO) classification, adult-type diffuse gliomas represent the most common primary malignant brain neoplasms in the clinical practice of neuro-oncology and neurosurgery [6], and they are consequently those with a greater economic and social impact. In particular, by evaluating mutations in isocitrate dehydrogenases (IDH) 1 and 2, ATRX chromatin remodeler, tumor protein p53 (TP53), telomerase reverse transcriptase (TERT), codeletion in 1p/19q, homozygous deletion in cyclin-dependent kinase inhibitors 2A and 2B (CDKN2A−B), gain in epidermal growth factor receptor (EGFR) and/or chromosome 7 and loss of chromosome 10 (7+/10−), adult-type diffuse gliomas can be classified in astrocytoma (IDH-mutant), oligodendroglioma (IDH-mutant and 1p/19q-codeleted), and glioblastoma (IDH-wildtype) [6]. These molecular profiles and peculiar morphological characteristics are able to define prognostic aspects, survival, and diagnosis [6]. Considering the great heterogeneity of these tumors, the identification of specific biomarkers for low- and high-grade gliomas is of great importance, especially for the development of potential novel chemotherapeutic agents. Indeed, the understanding of the specific involvement of phospholipases in these tumors makes this field of research extremely interesting for the scientific and health community, as it could pave the way for novel therapeutic targets for pharmacological interventions.

## 2. Phospholipases Characteristics

Phospholipases can be classified into four major classes according to the type of reaction that they catalyze [1,5]. Phospholipase A1 (PLA_1_), phospholipase A2 (PLA_2_), and phospholipase B (PLB) represent acylhydrolases, being able to preferentially cleave the acyl ester bond at the *sn-*1 acyl chain, the *sn-*2 acyl chain, and both the positions of PLs, respectively, whereas phospholipase C (PLC) and phospholipase D (PLD) constitute phosphodiesterases [7]. In particular, PLCs are responsible for hydrolyzing phosphatidylinositol 4,5-bisphosphate (PtdIns(4,5)P_2_) to generate the two essential intracellular second-messenger diacylglycerol (DAG) and inositol 1,4,5-trisphosphate (IP_3_) [8], leading to the activation of protein kinase C (PKCs) and the release of calcium ions (Ca^2+^) from intracellular stores, respectively [9]. On the other hand, PLDs hydrolyze the distal phosphodiester bond of PLs, releasing phosphatidic acid (PA) and the corresponding head-group [7]. According to the Human Protein Atlas (www.proteinatlas.org, accessed on 1 April 2023), phospholipase B (PLB) expression is limited prevalently to the distal intestine [10], and it is represented by low human brain regional specificity. For this reason, this review is focused on the other three phospholipases, which are abundant in the human brain [1]. 

Each family of phospholipase is characterized by different isoenzymes with specific functions, domains, and regulatory mechanisms of actions [11].

PLA: The PLA_1_ family is ubiquitously expressed in several eukaryotic organisms, such as yeast, plants, and mammals [5]. The mammalian family includes three members: p125/Sec23ip (iPLA1β), KIAA0725p/DDHD2 (iPLA1γ), and phosphatidic acid-preferring phospholipase A1 (PA-PIA1)/DDHD1 (iPLA1α). These isoforms are all characterized by the DDHD domain that is involved in their binding to specific organelle membranes, indispensable for their activity [12]. As shown by Joensuu et al., PLA_1_ enzymes are expressed in human brain with distinctive subcellular localization, but their physiological functions remain still largely unknown [1]. On the other hand, the enzymes belonging to the PLA_2_ family are categorized in six main classes, and their function is abundantly studied: secretory phospholipase A2 (sPLA_2_), cytosolic phospholipase A2 (cPLA_2_), calcium-independent phospholipase A2 (iPLA_2_), platelet-activating factor acetylhydrolases (PAF-AH PLA_2_), lysosomal phospholipase A2 (LPLA_2_), and adipose-specific phospholipase A2 (adPLA_2_). 

PLC: Phosphatidylinositol-specific PLC (PI-PLC) present in mammals are represented by 13 different isoforms split between 6 subfamilies, PLC β (1,4), PLC δ (1,2 and 4), PLC γ (1,2), PLC ε, PLC ζ, and the most recently discovered PLC η (1,2) isoform [8]. All reported PLCs bear conserved domains such as the X and Y catalytic domains, the pleckstrin homology (PH) domain, the EF-hand (EF-H) domain, and the PKC homology (C2) domain, and each of these regions has specific functional roles [13]: the PH domain binds to PtdIns(4,5)P_2_, and it is essential for the membrane linkage, the EF-H domain plays scaffolding roles in supporting guanosine triphosphate (GTP) hydrolysis upon the G-protein-coupled receptor (GPCR) binding, X and Y are catalytic domains in the form of a distorted triose-phosphate isomerase (TIM) barrel with a highly disordered and flexible intervening linker region, and C2 participates in intra- and inter-molecular signaling processes [8]. Some of the regulatory domains are uniquely distributed in PLC subtypes, and this may explain their distinct activities and their differential distributions in tissues [13,14]. Moreover, it has been demonstrated that PLCs can be placed across various cellular compartments, depending on the localization of their substrate [15]. Notably, a separate nuclear phosphoinositide metabolism, distinct from the cytoplasmic one, is today well recognized and studied by the scientific community, placing it at the center of many pathologies, including brain and tumor disorders [16,17]. Several evidences confirm the implication of nuclear PLC signaling in different tumor systems, especially in myelodysplastic syndromes [18,19], but also in breast cancer, melanoma, and pancreatic and colorectal cancers [8]. Moreover, many pieces of evidence indicate that some nuclear PLC isoforms can control endocannabinoid neuronal excitability and also the development of normal cortical circuitry, pointed out as key enzymes and targets in many brain processes [20].

PLD: Phospholipase D is ubiquitously expressed in bacteria and eukaryotes, and up to now, six putative PLD genes have been classified in humans, including PLD_1_ and PLD_2_ which represent the isoforms with the most well-known enzymatic role [21]. These isoforms contain the PH-regulating domain and the lipid-binding Phox consensus sequence (PX) domain, as well as an acidic PtdIns(4,5)P_2_-binding motif that is involved in the enzyme’s subcellular localization [1]. PLDs are controlled by a wide range of intracellular stimuli including phosphoinositides, small GTP-binding proteins, and PKCs [1]. It has been reported the pivotal involvement of PLDs’ reaction products (PA and other lipid mediators) in the regulation of actin cytoskeleton rearrangements, differentiation, and migration [5]. 

## 3. Phospholipase Signaling

Phospholipase isoforms belonging to different families can share molecules from common pathways. These enzymes can be activated by different signaling molecules, including hormones, growth factors (platelet-derived growth factor (PDGF), fibroblast growth factor (FGF), epidermal growth factor (EGF), insulin-like growth factor (IGF), and vascular endothelial growth factor (VEGF)), and lipids such as lysophosphatidic acid (LPA) and sphingosine 1-phosphate (SIP), and these stimuli can act on both receptor tyrosine kinases (RTKs) and GPCRs [22]. 

For instance, PLCβ and PLCγ isoenzymes are firstly actioned by extracellular stimuli, while the activity of PLCδ and PLCη, also known as secondary PLCs, can be enhanced by intracellular calcium mobilization [8]. It has been demonstrated that PLCβ isoforms are generally activated by the Gαq and Gβγ subunits of heterotrimeric G proteins, while PLCγ isoforms act through the phosphorylation mediated by the binding of their Src Homology 2 (SH2) domain to phosphorylated tyrosine residues of activated and non-activated RTKs. Most of the tyrosine kinases involved in the activation of PLCγ are members of the growth factor receptor, and sometimes, PLCγ activation can depend also on phosphoinositide 3-kinases (PI3K). As already mentioned, PLCs generate two essential intracellular second messengers, DAG and IP_3_, which lead to the activation of PKCs and the release of calcium ions from intracellular stores, respectively [23]. These events contribute to the stimulation of PLD, which, through PA generation, can stimulate different mechanisms including the phosphorylation of the extracellular signal-regulated kinase (ERK) [3,24]. This action, together with calcium binding, contributes to the activation of cPLA_2_. On the other hand, iPLA_2_ is generally phosphorylated by PKC [3]. 

It has been demonstrated that this hierarchical and intricate signaling network regulates several important biological processes implicated in cancer. For instance, through different lipid mediators, phospholipases can transmit downstream signals involved in tumorigenesis processes and aggressiveness, including cell proliferation, invasion, metastasis, growth, and angiogenesis [3]. 

## 4. Phospholipases in Brain Tumors

In the last years, phospholipases have attracted considerable interest in neuroscientific and nervous system disorders’ fields, and many studies, which are further explored below, have suggested their fundamental role in brain tumor development and progression [1,25]. 

### 4.1. PLA_2_ Function and Involvement in Brain Tumors

Although PLA_2_ role in the nervous system is not well defined yet, especially due to the great complexity in the expression of PLA_2_ groups in this system, its implication in membrane remodeling, exocytosis, and neurotransmitter release can explain the involvement in some brain lesions and in neurodegenerative diseases such as Alzheimer’s disease [26]. Consequently, it is clear the relevance of this molecule, especially the secreted isoform sPLA_2_, in the mechanisms of inflammation and the fact that it has been proposed as a marker for some of these events [27]. Moreover, it has been recently demonstrated the link between this inflammatory protein with astrocytoma’s proliferation via the activation of ERK- and PKC-driven signaling pathways [27]. These events are fundamental signaling mediators in many tumor types and contribute to aggravating the prognosis of tumors in inflammatory microenvironments. 

In addition to the secreted form, the cytosolic phospholipase A_2_ appears to play a key role in the pathogenesis of some brain tumors and particularly in glioblastoma. Nowadays, targeting multiple pathways implicated in glioblastoma metabolism is considered an increasingly effective therapeutic strategy, and cPLA_2_ has long been considered in the context of metabolic reprogramming, which is recognized as a hallmark of cancer [28]. It has been demonstrated that lipid remodeling promoted by cPLA_2_ and polymerase I and transcript release factor (PTRF), whose increase is generally associated with a worse prognosis in glioma patients [29], leads to tumor proliferation and glioblastoma survival by enhancing endocytosis and by promoting mitochondria energy metabolism and adenosine triphosphate (ATP) release [30]. It is well-known that ATP gives support for all the cell biological functions and consequently can affect the tumor cell proliferation and also the tumor cell microenvironment. Another study suggested cPLA_2_ as a potential therapeutic target for personalized therapies in glioblastoma patients, demonstrating that blood exosomes-based targeted delivery of cPLA_2_ siRNA and metformin, selectively targeted the tumor energy metabolism obtaining antitumor effects [31].

However, the role of cPLA_2_ is highly controversial, and some studies also elucidated its implication via its phosphorylation in the anti-tumor effects of temozolomide (TMZ), the alkylating agent used as the first line of treatment for glioblastoma [32]. In particular, it was shown that TMZ-induced phosphorylation of cPLA_2_ is connected to cell growth suppression of glioblastoma cells, except for cells with high expression of O-6-methylguanine-DNA methyltransferase (MGMT), which removes methyl groups transferred to DNA by TMZ [33], leading the cells to chemoresistance against the alkylating agent. Consequently, cPLA_2_ phosphorylation may be caused by TMZ-induced DNA alkylation, and drugs that decrease the phosphorylated form of cPLA_2_ might mitigate the anti-tumor effect of TMZ itself. Moreover, other findings proved that arachidonic acid generated by phosphorylated cPLA_2_ has a cytotoxic effect and decreases tumor cells’ progression, through reactive oxygen species (ROS) generation and through the increase of cytoplasmic calcium influx [34], in different types of brain cancer including neuroblastoma, glioma, and retinoblastoma [35]. These findings underline the importance of investigating these enzymes and the pathways connected to them more in detail, in order to clarify their effective roles in the pathogenesis of these heterogeneous tumors.

### 4.2. PLCs’ Potential Role in the Aggressiveness of Low- and High-Grade Gliomas

It has been recently demonstrated the potential involvement of some of the PLC family members in glioblastoma onset and aggressiveness [36,37]. Different studies have already shown the significance of lipid signaling and PLCs in the regulation of different cellular mechanisms such as cell proliferation, differentiation, migration, and the cell cycle, and in many physio-pathological brain processes [16,25]. 

The most frequently characterized PLC isoforms in the brain are PLCβ with all its four isoforms and PLCγ1 [25]. They are differentially distributed, suggesting a specific role for each subtype in different brain areas and, therefore, specific consequences following their modulations. For instance, it has been documented the involvement of PLCs in various brain disorders including epilepsy, movement and behavior disorders, neurodegenerative diseases, and also high-grade gliomas [25]. A 2016 in silico study showed PLCβ1 as a potential prognostic factor and a candidate signature gene for specific subtypes of glioblastoma [38]. In particular, it was demonstrated that PLCβ1 gene expression was inversely correlated with the pathological grade of gliomas and that patients with intermediate PLCβ1 expression had a higher chance of survival than the PLCβ1-downregulated group [38]. Moreover, it has already been demonstrated in the literature the possible involvement of PLCγ1 in glioblastoma progression and aggressiveness, probably taking into account its strategic position at a convergence point of several signaling pathways, including growth factor receptor signaling and adhesion receptor signaling for cell spreading, invasion, and migration [39]. Indeed, past studies have shown a link between PLCγ1 inhibition and the arrest of glioma cell motility of fetal rat brain aggregates and the reduction of cell invasion abilities following its downregulation, suggesting a possible anti-invasive therapeutic strategy for glioblastoma [40]. More recently, the topic was further explored, demonstrating a better understanding that the molecular mechanisms driving glioblastoma transformation could be necessary to uncover novel therapeutic strategies. Recent findings highlighted an effective role of PLCβ1 in supporting a less aggressive phenotype of the tumor, demonstrating that its gene was relatively less expressed in glioblastoma patients compared to in their healthy/low-grade counterparts [36]. Moreover, PLCβ1 downregulation in both immortalized and primary cell lines is related to increased cell migration, invasion, proliferation, cell survival, and the upregulation of mesenchymal markers and matrix metalloproteinases (MMPs) [36]. On the other hand, a translational study conducted by the same research team identified and confirmed PLCγ1 as a key enzyme for glioblastoma progression [37]. Data collected on patients’ biopsies and engineered cell models suggested a strong connection between PLCγ1 expression level and the acquisition of a more aggressive cancer phenotype [37]. In particular, it was demonstrated that PLCγ1 gene expression was higher in glioblastoma patients’ tissue samples compared to in healthy controls and that PLCγ1 downregulation in in vitro models led to a reduction in cell migration and invasion abilities. Instead, the opposite trend was highlighted following PLCγ1 overexpression [37]. Taken together, these data unearthed the importance of further investigating PLCs as potential prognostic biomarkers and targets in the development of new therapeutic strategies for glioblastoma. 

As these recent discoveries demonstrate, even though PLCs are clearly implicated in cancer development and progression, their specific regulatory functions are sometimes elusive and controversial. Indeed, these enzymes could trigger in a different way the same molecular mechanisms, representing promising prognostic or therapeutic targets in cancer therapies due to their influential roles in cell proliferation, migration, growth, and survival.

The hypothesis that PLCγ1 could work as a potential therapeutic target for gliomas is also supported by a recent work that claims an association between PLCγ1 overexpression, tumor progression, and negative prognosis in patients characterized by IDH-wildtype lower-grade gliomas (LGGs) [41]. These low-grade brain tumors include oligodendrogliomas and astrocytomas, which account for about a quarter of all gliomas [42]. The results of this research showed that PLCγ1 silencing in IDH-wildtype LGG cell lines dramatically affects their progression, migration, and invasiveness and that the PLC-targeted drug significantly suppresses the tumor growth of both IDH-wildtype in vitro and mouse models [41]. 

Recently, the involvement of PLCε in the aggressiveness of gliomas was also demonstrated for the first time [43]. Particularly, it was shown an abundant expression of this PLC in a particular tumor population represented by glioma stem cells (GSCs), which are responsible for tumor aggressiveness, resistance, and recurrence in gliomas [44]. In the last years, there has been an increasingly evident interest in the development of therapies targeting GSCs, with the aim of improving the prognosis of patients with gliomas, and especially glioblastoma. Given their ability to differentiate into different lineages, GSCs retain an important therapeutic potential for regenerative medicine, and the understanding of the signaling pathways involved in their pluripotency maintenance and differentiation has a great importance in order to comprehend these mechanisms and develop novel therapeutic strategies [45]. Therefore, PLCε was suggested, for the first time, as a promising anti-GSCs therapeutic target, being involved in GSCs survival, maintenance, and stemness, as its in vitro inhibition leads to cell survival suppression and cell death [43]. Moreover, mouse models transplanted with PLCε knockdown in GSCs were characterized by longer survival compared to in the controls [43]. All together, these recent studies highlight an important milestone in GSCs research, focusing attention on PLCs as potential prognostic biomarkers and targets in the development of new therapeutic strategies for gliomas. 

Recent findings have also associated the enzymatic activities of PLC and PLD, with sphingomyelinases (SMase) and sphingosine-1-phosphate (S1P) signaling, which turns out to be involved in the mechanisms of proliferation and invasion of glioblastoma [46]. In particular, it has been demonstrated that tumor cells are able to evade apoptosis mediated by the main therapeutic agents in use, thanks to the conversion of ceramides produced by hydrolysis of sphingomyelins into S1P [46]. Therefore, one of the main ways that S1P appears to act within the cell is through synergy with phospholipase-mediated signaling pathways, leading to tumor progression and metastasis. 

### 4.3. PLD Involvement in Glioblastoma

Different reports indicated that the lipid-metabolizing enzyme phospholipase D and its associated signaling pathways may be clinically important therapeutic targets for glioblastoma. Among various considerations, it has been evidenced PLD_2_ as a key regulator of pro-survival RAC(Rho family)-alpha serine/threonine-protein kinase (Akt) in glioblastoma [47]. In particular, it has been established that phosphatidic acid, the product of the PLD reaction, is fundamental for the membrane recruitment and activation of Akt. The latter is a downstream kinase in the RTK/phosphatase and tensin homolog (PTEN)/PI3K pathways, which is largely mutated in the majority of glioblastoma patients [48]. In this contest, Akt represents a key signaling point which allows for the amplification of growth signals, thus making Akt inhibition an attractive target for tumor therapy [48]. This study demonstrated, for the first time, that the inhibition of PLD_2_ activity and the following Akt behavior lead to the decline of glioblastoma cell viability through the inhibition of the autophagic flux [48]. 

Recent findings also highlighted the interesting role of the PLD_1_ isoform, in the physiology of GSCs and especially in recurrent glioblastoma [49]. In particular, it has been demonstrated a positive correlation between PLD_1_ expression and the level of the transmembrane protein cluster of differentiation 44 (CD44), which is a well-recognized marker of the mesenchymal subtype of glioblastoma and a surface marker of cancer stem-like cells and consequently a sign of worse prognosis and a predictor of radio-resistance [50]. This paper suggested PLD_1_ inhibition as a potential therapeutic strategy for glioblastoma, showing this approach as more beneficial when used in combination with the administration of TMZ, compared to treatment with TMZ alone [49]. In detail, PLD_1_ inhibition sensitizes GSCs to the alkylating agent and reduces glioblastoma tumorigenesis, pointing out the aim of the current research on glioblastoma, which is focused on any novel therapy that could address GSC-driven tumor recurrence and resistance to TMZ.

## 5. Conclusions

This review shows that phospholipases belonging to different families (PLA, PLC, and PLD) play a pivotal role in the regulation of cell signaling in brain tumors and are consequently implicated in the pathogenetic and clinical evolution of low- and high-grade gliomas as tumor suppressors or oncogene, thus proving to be attractive prognostic biomarkers or therapeutic targets, respectively, for these heterogeneous tumors with no real effective cure. (The main correlations between phospholipases’ gene expression and tumor phenotype are shown in Table 1). 

Indeed, despite the currently existing therapeutic approaches for gliomas of various degree of aggression, the prognosis and existing therapies have remained unchanged for years [51]. Consequently, in order to improve cancer research and work towards overall positive clinical outcomes, it is necessary to elucidate the complex synergy of various molecules involved in phospholipases-mediated mechanisms and the resultant physiological implications. Indeed, even though phospholipases and their mediators are clearly implicated in the development of cancer [3], their specific regulatory functions are sometimes complex and controversial, which underlines the necessity to further investigate the diverse isoforms, separately and in parallel. Indeed, these enzymes could regulate in a different way the same molecular mechanisms, representing promising prognostic or therapeutic targets in cancer therapies due to their key roles in cell proliferation, migration, growth, and survival and in the main mechanisms involved in metabolic reprogramming. It is well-known that tumor cells are able to modify their metabolism to bear their cell growth and adapt to challenging microenvironments [52]. For this reason, the various mechanisms implication in the reprogramming of cell metabolism can be considered for therapeutic targeting of cancer. Indeed, phospholipid metabolism is largely altered in different types of neoplasms including gliomas [53]. However, the influence and functions of phospholipid remodeling in tumor cells are still poorly understood. The complete understanding of these events could allow the correlation between tumor pathological mechanisms and the identification of future useful diagnostic and prognostic biomarkers in gliomas.

## Figures and Tables

**Figure 1 biomolecules-13-00798-f001:**
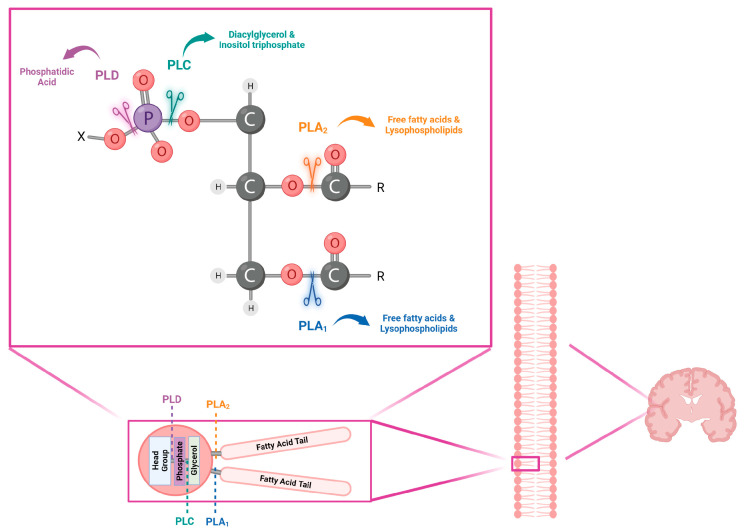
Graphic representation of phospholipases’ cleavage sites on phospholipids, the most abundant species contributing to the biological membranes of nervous cells, and their respective products. Each phospholipase with a relative cleavage site (shown with scissors) and reaction products is represented by a specific color (from the left to the right of the figure: PLD in purple, PLC in green, PLA_2_ in orange, and PLA_1_ in blue). X: functional group. O: oxygen; P: phosphorus; C: carbon; H: hydrogen; R: fatty acid tails. Phospholipase B is not shown in the figure, as it is not the subject of study in this paper.

**Table 1 biomolecules-13-00798-t001:** Aberrant expression and consequent clinical potential of phospholipases in brain tumors.

Phospholipase	Brain Tumor Type	Expression	Clinical Potential	References
sPLA_2_	Astrocytoma	Increased	Therapeutic target	[27]
cPLA_2_	Glioblastoma	Increased	Therapeutic target	[30,31]
PLCβ1	Glioblastoma	Decreased	Prognostic biomarker	[36,38]
PLCγ1	Glioblastoma and LGG (*IDH-wt*)	Increased	Therapeutic target	[37,40,41]
PLCε	Glioblastoma	Increased	Therapeutic target	[43]
PLD_2_	Glioblastoma	Increased	Therapeutic target	[47]
PLD_1_	Glioblastoma	Increased	Therapeutic target	[49]

## Data Availability

No new data were created or analyzed in this study. An exception is represented by the data related to the expression of phospholipase B. These data can be found here: Human Protein Atlas (www.proteinatlas.org, accessed on 1 April 2023).

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
