# Peer review of "Phospholipases in Gliomas: Current Knowledge and Future Perspectives from Bench to Bedside"

_biomolecules, 2023, doi:10.3390/biom13050798_

Round 1
Reviewer 1 Report
This is a very well written an appropriate review about current knowledge about the role of phospholipases in brain tumor progression, focusing on low- and high- grade gliomas, representing promising prognostic or therapeutic targets in cancer. I have a minor comment : phosphatidylinositol 4,5-bisphosphate on lane 86
- Review is clear.
- No similar review I am aware of
- Refs are OK
- Citations relevant
- No relevant citation omitted
- Conclusions relevant
- Regarding the graphical abstract. It could be improved. What represents the colour code ? A legend could be added to the graph (or to the main text) to emplasize these heterogenous tumours which could be very much different from one patient to the other. Actually I very much like Table 1 with a summary in which refs are directly available to appreciate the limitation but also the advantage of measuring the expression of the different phospholipases.
Reviewer 2 Report
This is a review focused on phospholipases and its impact on gliomas. The authors begin by describing the different phospholipases including their expression and signaling. The central part of this review is the description of phospholipases and brain tumors. The content of this manuscript is good; however, several points must be addressed before it can be considered for publication:
1. The authors mention on page 2, line 59, that phospholipases are categorized into 4 classes (A-D) according to figure 1. However, figure 1, which shows the various sites of action of these enzymes, does not specify which of these is A, B, C or D, so the figure must be improved, and the figure caption must contain a detailed description of it.
2. Authors should add a representative figure of the mechanism by which the phospholipases impact the glioma.
3. Authors should add a section describing the characteristics of gliomas, which means whether they are considered high grade or low grade or whether they express IDH.
4. Section 4.1 needs to be improved, as the information is in isolated paragraphs that do not have a clear sequence.
Reviewer 3 Report
The manuscript by Marvi et al., “Phospholipases in gliomas: current knowledge and future perspectives from bench to bedside” provides as concise review of the biological functions of phospholipases, their suspected role in tumor proliferation and growth, and their potential as targets for therapeutic purposes. The manuscript is generally well written and can be published after these minor issues are addressed.
General:
There are several acronyms that are used but which are never defined, these abbreviations should be defined after their first use. Furthermore, there are several abbreviations that are provided (MMPs, TIM, SMase, etc.) but then never used again in the manuscript. These abbreviations are not necessary and should be removed. A list of abbreviations would be useful.
There needs to be a deeper discussion on some of the examples that are provided. For some references that are provided, only a single sentence of discussion is given. Where the author's own subject matter is the focus, the information is in depth and well summarized.
The following minor issues need to be addressed:
Line 55 - The phrase “These latter are” should be replaced by “Phospholipases are” or something similar.
Line 55 - Change has to have.
Line 58 - A better word for peculiar may be particular. The use of the word peculiar occur several times in the manuscript and there are likely more appropriate terms to use. When I read the word peculiar, I think that something is unusual.
Line 58 - “As a matter of fact”, see comment above regarding the use of this phrase.
Line 73 - Change originated to the present tense originate. A little background information of these different tumor types would be beneficial.
Line 92 - Correct...it il represented.
Line 142 - Correct lipids as to lipids such as.
Line 167 - If many studies have suggested, references to these previous many studies need to be provided.
Line 189 - Correct cPLA2 to cPLA2 to be consistent.
Line 206 - Other studies suggested implies more than one study. Correct the grammar to refer to the single provided reference, or provide additional references to support this claim. A discussion of these additional references will need to be included.
Line 277 - This sentence needs to be rewritten; it reads as if it is referring to the reference (Ref. 42) in the previous sentence which is a review by some of the authors of this current manuscript, when it appears to refer to the provided reference at the end of the sentence Ref. 40.
The quality of the English with which this manuscript was written is generally quite good. However, there are phrases that can be removed as they add little to the intended meaning of the sentence and the overall impression of the manuscript.
There are many sentences that begin with the phrase “As a matter of fact”, this phrase is used much too often, and does not contribute any meaning the sentence in which it is present. In these instances, being more direct is a better approach.
In several places throughout the manuscript, there are many phrases that are awkward. For example: line 212 - "Already in the past", line 226 - "it has already been evidenced". There are more clear ways to convey the intended meaning of these particular phrases.
Round 2
Reviewer 2 Report
Authors took into consideration the comments and the manuscript improved.